# Fine Mapping and Cloning of a *qRA2* Affect the Ratooning Ability in Rice (*Oryza sativa* L.)

**DOI:** 10.3390/ijms24020967

**Published:** 2023-01-04

**Authors:** Niqing He, Fenghuang Huang, Dewei Yang

**Affiliations:** Rice Research Institute, Fujian Academy of Agricultural Sciences, Fujian High Quality Rice Research and Development Center, Fuzhou 350018, China

**Keywords:** rice (*Oryza sativa* L. subsp. *indica*), gene mapping, gene cloning, ratooning ability, near-isogenic lines

## Abstract

Ratooning ability is a key factor that influences the ratoon rice yield in areas where light and temperature are not sufficient for second-season rice. Near-isogenic lines (NILs) are the most powerful tools for the detection and precise mapping of quantitative trait loci (QTLs). In this study, using 176 NILs, we identified a novel QTL for ratooning ability in NIL128. First, we mapped the QTL between the markers Indel12-29 and Indel12-31, which encompass a region of 233 kb. The rice genome annotation indicated the existence of three candidate genes in this region that may be related to ratooning ability. Through gene prediction and cDNA sequencing, we speculated that the target gene of ratooning ability is *LOC_Os02g51930* which encodes cytokinin glucosyl transferases (CGTs), hereafter named *qRA2*. Further analysis showed that *qra2* was a 1-bp substitution in the first exon in NIL128, which resulted in the premature termination of *qRA2*. The results of the knockdown experiment showed that the Jiafuzhan knockdown mutants exhibited the ratooning ability phenotype of NIL128. Interestingly, the *qRA2* gene was found to improve ratooning ability without affecting major agronomic traits. These results will help us better understand the genetic basis of rice ratooning ability and provide a valuable gene resource for breeding strong ratoon rice varieties.

## 1. Introduction

Rice (*Oryza sativa* L.) is one of the major staple foods worldwide, feeding more than 65 percent of China’s population [1]. Maintaining a sustained increase in rice production is particularly important to meet the needs of a growing population, thereby ensuring national food security [2]. However, due to the rapid growth of the world’s population, rice production will have to increase significantly in the next half century to meet the needs of world population growth. Obviously, this increase in rice production requires higher yields and more frequent harvests on existing land [3]. This is mainly because it is difficult to increase rice yields further by expanding the area under cultivation [4].

Ratoon rice, related to ratooning ability, is a second crop formed by the regeneration of tillers from the remaining nodal buds of the stubble after the first harvest, thus increasing the total yield by increasing the frequency of the harvests [5,6]. Compared with other rice planting modes in China, the annual production yield of ratoon rice is significantly higher than that of single-season rice, while the net economic benefit is higher than that of double-season rice [7]. Therefore, the rapid development of regrown rice in China and around the world, both in applied research and basic research, has become a worldwide research hotspot in recent years [8,9,10].

To date, few studies have been reported on the ratooning ability of rice, and its genetic basis is still unclear. Six QTLs for ratooning ability were detected in a doubled haploid population [11]. Using advanced backcross lines, a QTL called *qRAT5* was detected between InDel3 and RM249 on chromosome five [12]. One QTL for ratooning ability was identified using an SSR marker [13]. Two QTLs for ratooning ability were identified by a recombinant inbred line population [14]. Three QTLs for ratooning ability were detected by a recombinant inbred line population [15]. Recently, a QTL named *qRA5* was located between 14.12 Mb and 14.65 Mb with a distance of 3.1 Mb using an introgression line population [16].

The genes encoding cytokinin glycosyl transferase (CGT) and their potential applications in crop yields have received little attention. Gene coding for a xylosyl transferase was first isolated from *Phaseolus lunatus* [17]. Li et al. [18] identified the first cytokinin glycosyltransferase Os6 from rice, and further research revealed that Os6 was expressed in both the vegetative and reproductive tissues of rice. Recent studies have shown that CGT is a key regulator of cytokinin homeostasis and has potential value for wheat genetic improvement [19].

Therefore, it is the most effective way to improving the yield of ratoon rice will require better understanding of the specific functions of ratooning ability-related genes. The strength of ratooning ability plays an important role in the stable and high-yield cultivation of ratoon rice, however, the identification and genetic mechanism of ratooning ability-related genes remain unclear. Consequently, further studies are needed to elucidate the genetic mechanism of ratoon rice formation, cultivate high-ratoon rice varieties, and promote the development of the ratoon rice industry by optimizing cultivation techniques. In this study, Hui1586, a japonica cultivar, was used as the donor parent, and an indica cultivar, Jiafuzhan, was used as the recipient parent. Ratooning ability was evaluated using 176 NILs. The gene *qRA2*, which affects the ratoon ability-related traits, was validated using NIL128, and its genetic background effect was evaluated. Furthermore, *qRA2* was fine mapped within 233 kb by a secondary population, and *LOC_Os02g51930* was identified as a candidate gene. Using clustered regularly interspaced short palindromic repeats/CRISPR-associated nuclease 9 (CRISPR/Cas9) genome editing technology, we knocked out the *qRA2* gene in the background of Jiafuzhan and observed that the knockout mutants exhibited the ratooning ability phenotype of NIL128. Interestingly, the *qRA2* gene was found to improve ratooning ability without affecting major agronomic traits. These results indicate that *qRA2* has good application prospects in ratoon rice breeding in the future.

## 2. Results

### 2.1. Identification and Analysis of QTLs for Ratooning Ability in the NILs

To evaluate the potential advantages of the NILs for QTL detection, phenotypic variations in ratooning ability were observed in 176 NILs, Jiafuzhan, and Hui1586 at the Fuzhou Experimental Station in July 2019 (Fuzhou, Fujian, China), and it was found that NIL128 exhibited a lower ratooning ability than Jiafuzhan (Figure 1). There was no significant difference between the ratooning ability of the remaining 175 NILs and that of Jiafuzhan.

We further analysed the reasons for the changes in the ratooning ability of NIL128. WinQTLCart 2.5 software was used to analyse the relevant QTLs. The results showed that the LOD score of the QTL for the ratooning ability of NIL128 was 28.26, and the explained phenotypic variance (R2) was 17.82%. Further analysis showed that the ratooning abilities of Jiafuzhan and Hui1586 were 137.5% and 52.3%, respectively, while that of NIL128 was 48.2%. According to the t-test, the differences in the ratooning ability of these strains reached a significant level (Table 1; Figure 1).

Phenotypic comparisons between NIL128 and Jiafuzhan were performed at the Fuzhou experimental station in June 2021 (Fuzhou, Fujian, China) (Table 1). Apart from the ratooning ability, the only two agronomic traits that showed differences were the seed set rate and yield per plant between the NIL128 line and the Jiafuzhan cultivar. However, there were no significant differences in the remaining agronomic traits (Table 1).

### 2.2. Genetic Analysis of NIL128 for Ratooning Ability

To conduct a genetic analysis of the ratooning ability of NIL128, NIL128 was hybridized with Jiafuzhan. The results showed that F_1_ hybridization exhibited the phenotype of Jiafuzhan of ratooning ability, and the F_2_ population phenotype was consistent with the Mendelian separation ratio (Table 2). Segregation between the Jiafuzhan and NIL128 phenotypes fit a 3:1 segregation ratio in the two F_2_ populations (χ^2^ = 0.134–0.456, *p* > 0.5). The results showed that the regeneration ability phenotype of NIL128 was inherited by invisible traits.

### 2.3. Analysis of the Genetic Background of NIL128

The genetic background of NIL128 homozygous material was analysed using 296 primers that showed polymorphism between Hui1586 and Jiafuzhan cultivars [20]. The results showed that five markers exhibited homozygous alleles of Hui 1586, including Ind1-3 on chromosome 1, RM7286 on chromosome 2, Ind3-18 on chromosome 3, RM566 on chromosome 9, and RM19 on chromosome 12. The remaining 291 markers showed the genetic backgrounds of the Jafuzhan variety (Appendix A). Therefore, we speculate that the NIL128 phenotype for ratooning ability may originate from these five loci.

### 2.4. Linkage Analysis of QTLs for Ratooning Ability in NIL128

To determine which gene causes the NIL128 phenotype, we located the QTL of NIL128 for ratooning ability based on five potential candidate sites. A total of 325 recessive plants from the F_2_ population were used for linkage analysis between five markers and the QTL. One of five SSR markers, RM3795, was found to be linked to the trait among the 325 F_2_ individuals.

### 2.5. Initial Localization of QTLs for Ratooning Ability

To initially locate the QTL, published markers were used around RM3795. The QTL for ratooning ability was located between the molecular markers RM1386 and RM7286 at a distance of 3.1 Mb by linkage labeling (Figure 2a), and each molecular marker corresponding to the physical distance obtained according to each molecular marker sequence.

### 2.6. Fine Mapping of QTLs for Ratooning Ability

To fine map, the QTL, fourteen polymorphic markers were selected from 42 newly developed molecular markers (Table 3). Six polymorphic markers were used to genotype all recombinant strains. The results showed that the QTL was located between the molecular markers Indle2-20 and Indle2-24 on chromosome 2 at a physical distance of 233 kb (Figure 2b and Table 3).

### 2.7. Candidate Genes in the 233-kb Region

According to the gene annotation database (http://rice.plantbiology.msu.edu/; http://www.tigr.org/ and accessed on 13 November 2020), there were fifty-one annotated genes in the 233-kb region (Figure 2c), and almost every gene had a corresponding full-length cDNA. After further analysis, we found that three genes were speculated to be related to ratooning ability: *LOC_Os02g51900*, *LOC_Os02g51910*, and *LOC_Os02g51930*. All three genes encode cytokinin-O-glucosyltransferase 2.

### 2.8. Sequence Analyses of the QTL for Ratooning Ability

To investigate which gene was responsible for the phenotype, we then sequenced three candidate genes of Jafuchian and NIL128. The results showed that a 2-bp substitution (493: G⇨T and 1297: T⇨G) was found in *LOC_Os02g51930* (Figure 2c), and there was no difference in the sequence of the remaining two genes. Therefore, we speculate that *qra2* was identified as having a weak ratooning ability in NIL128, while *LOC_Os02g51930* corresponds to *qRA2*.

Analysis of the structure of open reading fragments (ORFs) showed that *qRA2* had two exons; compared with the reference sequence Nipponbare (Nip), there was a base substitution (1297: T⇨G) at the second exon that caused the amino acid to change from C to G (Appendix A). Further analysis revealed that *qra2* was a 1-bp substitution (493: G⇨T) in the 493 rd base of the first exon, which resulted in the premature termination of *qRA2* (Appendix A).

### 2.9. qra2 Is Responsible for the Ratooning Ability Phenotype of NIL128

To confirm whether *qra2* is associated with ratooning ability, we investigated whether knocking out *qRA2* in Jiafuzhan results in the ratooning ability phenotype of NIL128. A sequence-specific guided RNA (sgRNA) knockout of the *qRA2* gene was designed using the CRISPR/Cas9 gene editing system. Two homozygous lines (*qPA2KO-line1* and *qPA2KO-line2*) were obtained from two independent events and confirmed by sequencing the insertion or deletion of their main carrying targets in December 2020 (Table 4).

In April 2021, *qRA2KO-line1*, *qRA2KO-line1*, Jiafuzhan, Hui1586, and NIL128 were planted at the Fuzhou Experimental Station. We then investigated the ratooning ability phenotypes of these two homozygous lines (*qPA2KO-line1* and *qPA2KO-line2*) in the first season and ratooning season and found that both lines showed the phenotype of NIL128 in the ratooning season. In addition, there was no significant difference between the two knockout lines (*qPA2KO-line1* and *qPA2KO-line2*) and Jiafuzhan in terms of the main agronomic characteristics, including plant height, panicle length, number of effective panicles, number of spikelets per panicle, seed setting rate, 1000-grain weight, grain length, grain width, and yield per plant (Figure 3 and Table 1). Therefore, we speculated that the loss of *qRA2* function was the main reason leading to the weak ratooning ability phenotype of NIL128, and *qRA2* may not affect the main agronomic traits of rice.

### 2.10. Tissue Expression Analysis of qRA2

To further understand the function of *qRA2*, quantitative reverse transcription-PCR (qRT−PCR) was used to detect the expression patterns of *qRA2* at different developmental stages of rice. The results showed that *qRA2* was expressed in all tissues tested here, including roots, shoots, and leaves of two- and four-week-old panicles of 0.5–1 cm, 1–3 cm, 3–5 cm, and 5–10 cm length, along with germinating and mature seeds and callus, however, it was predominantly expressed in the seed (germination and mature) and four-week-old shoots (Figure 4).

## 3. Discussion

### 3.1. qRA2, a New Gene Associated with Ratooning Ability

In areas where light and temperature are not suitable for the growth of second-season rice, it is important to expand the planting area of reproducing rice. However, the ratooning ability is a complex quantitative trait that is controlled by multiple genes and greatly influenced by the environment. NILs have been used in plant genetic studies, QTL isolation, and fine mapping of genome-wide target traits [21]. The main feature of the NIL population is that each NIL contains only one or more donor chromosome fragments, which has a unique advantage for QTL identification, and by masking background genetic noise, QTLs can be visualized as a single Mendelian factor. Using the population of NILs, several QTLs have been identified or cloned [22,23,24,25,26,27,28]. In this study, we constructed 176 NILs using Jiafuzhan as the genetic background. Using these 176 NILs, we mapped the ratooning ability QTL and designated it *qRA2*. Map-based cloning and knockout experiments confirmed that the weak ratooning ability of NIL128 was caused by the loss of CGT function.

CGT is significantly related to plant responses to internal and external environments. From the perspective of the application, progress has been made in controlling related agronomic traits by regulating the level of cytokinin within agronomic species [19]. It was found that in rice when the expression of glucose transferase was reduced by RNAi, both grain number/panicle number and grain size were increased [29]. In addition, in a study of wheat, it was found that among the five wheat varieties with different yields, the varieties with higher yields had a lower glucose transferase expression [30]. Recent studies have shown that CGT is a key regulator of cytokinin homeostasis and has potential value for wheat genetic improvement [19]. Our study revealed that NIL128, compared with Jiafuzhan, had a lower seed setting rate, resulting in a decrease in yield. However, we did not find any effect on the yield in the two knockout lines of Jiafuzhan (Table 1). We speculated that this was probably due to the different genetic backgrounds between NIL128 and the knockout lines, and there were five differences in genetic background analysis between NIL128 and Jiafuzhan (Appendix A). Although our study revealed that knockout of CGT did not affect rice yield traits, it remains to be further verified whether changes in *qRA2* expression levels and overexpression affect rice yield-related traits.

### 3.2. The Application Prospect of the qRA2 Gene in Rice Breeding

Breeding is a combination of science and art. In the breeding process, we need to accumulate good genes together. However, we have some traits that are difficult to evaluate directly in the field and require molecular techniques, such as quality, resistance, and ratooning ability. At the same time, some genes are also associated with undesirable agronomic traits, so it is difficult to use them directly. Studies have shown that some resistance *R* genes are associated with undesirable agronomic traits, often at the cost of “sacrificing” yield, making it impossible to achieve both yield and resistance [31,32]. In this study, our data indicated that the *qRA2* gene not only enhanced the ratooning ability of rice but also did not affect the important agronomic traits of rice (Table 1 and Figure 3). Meanwhile, the identification of QTLs related to ratooning ability and the introduction of QTLs into excellent rice varieties are effective ways to improve rice viability. Together, the results indicate that *qRA2* has good application prospects for breeding strong ratooning ability varieties in the future. For example, to improve the ratooning ability of rice varieties, *qRA2* can be transferred into different rice materials. 

## 4. Materials and Methods

### 4.1. Plant Material

The indica rice cultivar Jiafuzhan and the japonica rice cultivar Hui1586 were preserved at the Rice Research Institute, Fujian Academy of Agricultural Sciences, China. In the early stage, 176 NILs were constructed with the indica cultivar Jiafuzhan as the acceptor and the japonica cultivar Hui1586 as the donor [20].

### 4.2. Investigation of Ratooning Ability

In this study, the final ratooning rate was used to express the ratooning ability, and the final ratooning rate (%) = the number of effective ears of reproducible rice/the number of effective ears of full-season rice × 100. Five effective panicles were randomly sampled in the middle of each plot, and the average of three replicates was used as the investigation result [33].

### 4.3. QTL Analysis of Ratooning Ability

In this study, WinQTLCart 2.5 software (http://statgen.ncsu.edu/qtlcart/WQTLCart.htm) [34] was used to achieve composite interval mapping (CIM) for QTL detection [35]. The confidence interval was defined as the 1-LOD reduction region near the peak LOD site of the identified QTL, and LOD ≥ 2.5 was used as the threshold for QTL identification.

### 4.4. Identification of QTLs for Ratooning Ability

In April 2019, Jiafuzhan, Hui 1586, and 176 NILs were planted in paddy fields under natural conditions at the experimental farm of the Fujian Academy of Agricultural Sciences (Fuzhou, Fujian, China). For parents and each NIL, a total of 64 plants were planted in 6 rows. All plants were grown using standard commercial methods, with row spacings of 13.3 cm and 26.4 cm, and field management techniques followed conventional planting methods.

Five plants were selected from the middle of each plot to study the characteristics of ratooning ability, and QTLs were determined according to the significance of differences between parents. In addition, the related agronomic characteristics, mainly plant height, panicle length, number of effective panicles, number of spikelets per panicle, seed setting rate, 1000-grain weight, grain length, grain width, and yield per plant, were measured at the maturity stage.

### 4.5. Construction of the Mapping Population

NIL128 was hybridized with Jafuzhan to construct a localization population at the Fuzhou Experimental Station in July 2019 (Fuzhou, Fujian, China), and the F_1_ plants were grown at the Sanya Experimental Station in November 2019 (Sanya, Hainan, China). The F_2_ localization population was formed by F_1_ self-crossing, and all F_2_ plants were planted at the Fuzhou Experimental Station in April 2020 (Fuzhou, Fujian, China). The primary linkage of ratooning ability QTLs was obtained from 45 recessive plants in the F_2_ population, and 325 recessive plants in the F_2_ population were selected for fine localization.

### 4.6. Molecular Marker Amplification and Detection

DNA was extracted from rice leaves by the CTAB method and modified appropriately [36]. For PCR amplification, every 20-μL reaction mixture contained 30 ng DNA (2-μL), 0.4 μM of each primer (1-μL), 2× Es Tag MasterMix (Dye) (10-μL), and 6-μL sterile deionized water [20]. The amplified PCR products were stained by 3% agarose gel electrophoresis with ethidium bromide staining as the marker [37].

### 4.7. Molecular Mapping of QTLs for Ratooning Ability

The bands of Hui 1586 and Jiafuzhan were denoted as 1 and 3, respectively, and the heterozygotes were denoted as 2. MAPMAKER version 3.0 software was used to analyse the association between QTL loci and molecular markers. MapDraw V2.1 was used to estimate genetic distance [38], and the linkage map construction method was the same as that previously reported [39]. The common primers used near RM3795 are references to this database (http://archive.gramene.org/markers/ and accessed on 12 August 2020), and Indel markers were designed on the basis of published rice genome sequences, and the polymorphisms between NIL128 and Jiafuzhan were predicted by comparing the sequences of Nipponbare (http://rgp.dna.affrc.go.jp/ and accessed on 1 September 2020) and 93-11 (http://rice.genomics.org.cn/ and accessed on 11 September 2020). Possible polymorphic primers were designed according to the different sequences of the target region sequences, and polymorphisms between NIL128 and Jiafuzhan were further detected.

In order to obtain each molecular marker in rice chromosome physical distance, we first opened the database (http://plants.ensembl.org and accessed on 3 September 2021), clicked the “BLAST” program, pasted the preprimer or postprimer sequence of primers on both sides of a marker into the box of “Enter the Query Sequence”, selected the species “Rice”, and clicked “Run” for retrieval. The specific position of the molecular marker on the physical map of rice can be obtained.

### 4.8. Physical Map Construction and Bioinformatics Analysis of QTLs for Ratooning Ability

The physical map of QTLs for ratooning ability was constructed based on the information of the International Rice Genome Sequencing Project (IRGSP, http://rgp.dna.affrc.go.jp/IRGSP/index.html and accessed on 9 October 2020). The clones were anchored with the target gene-linked markers, and sequence alignment was then carried out using the pairwise Basic Local Alignment Search Tool (http://www.ncbi.nlm.nih.gov/blast/bl2seq/b12.html and accessed on 9 October 2020). The candidate genes were predicted according to the existing sequence annotation database (http://rice.plantbiology.msu.edu/; http://www.tigr.org/ and accessed on 13 November 2020).

### 4.9. Targeted Mutagenesis of qRA2 in Rice with CRISPR/Cas9

We used the CRISPR plant database and website to design highly specific gRNA sequences at 267 bp on the first exon of the *qRA2* gene (Appendix A) [40]. Deletion and insertion of target genes were detected by PCR technology, and PCR amplification products were sequenced from the transgenic CRISPR-edited lines to determine the location and type of target gene mutations [41]. Primers for the CRISPR/Cas9 study are listed in Appendix A. Sequence primers for *qRA2* gene cDS amplification are shown in Appendix A.

## 5. Conclusions

In this study, a novel QTL for ratooning ability was mapped to a 233-kb locus region. Then, CRISPR/Cas9 knockout experiments confirmed that *qra2* is responsible for the ratooning ability phenotype of NIL128. Importantly, the *qRA2* gene, encoding CGT, was found to improve ratooning ability without affecting major agronomic traits, indicative of potential applications of *qRA2* in rice breeding programs. Taken together, our results will help us better understand the genetic basis of rice ratooning ability and provide a valuable gene resource for breeding strong ratoon rice varieties.

## Figures and Tables

**Figure 1 ijms-24-00967-f001:**
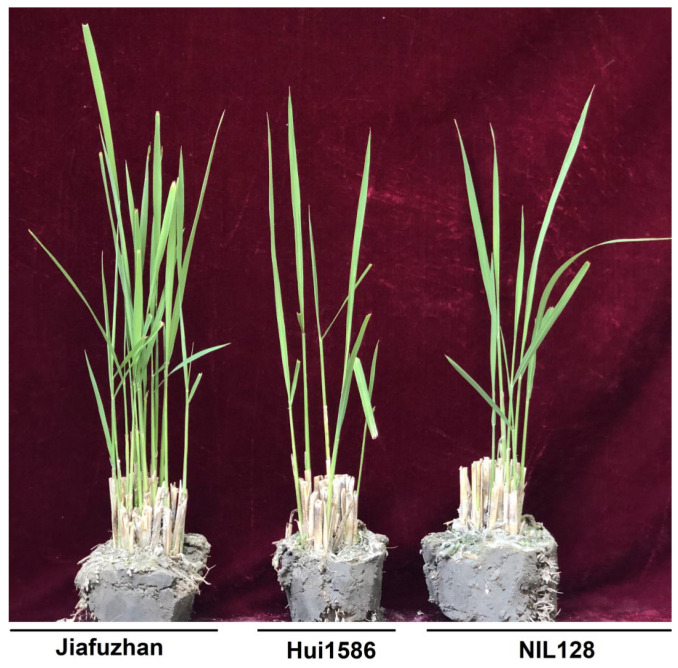
Ratooning ability comparison of Jiafuzhan, Hui1586, and NIL128. The phenotypes of Jiafuzhan, Hui1586, and NIL128 during the ratooning period are shown.

**Figure 2 ijms-24-00967-f002:**
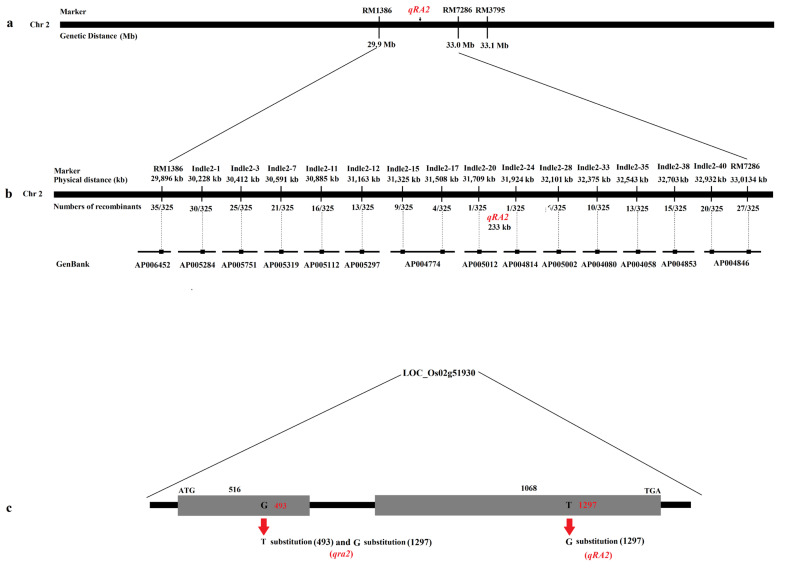
Physical maps and structural comparison of *qRA2*. (**a**): Primary mapping of *qRA2*. The QTL was mapped to the region between markers RM1386 and RM7286. (**b**): Further mapping of *qRA2*, which was localized at a 233-kb region between the markers Indel2-20 and Indel2-24, and the recombinant number between the markers. In addition, the GenBack was marked below each primer marker. (**c**): *qRA2* has two exons; compared with Nip, *qRA2* showed a 1-bp substitution in the second exon, and *qra2* showed a 1-bp substitution in the first and second exons.

**Figure 3 ijms-24-00967-f003:**
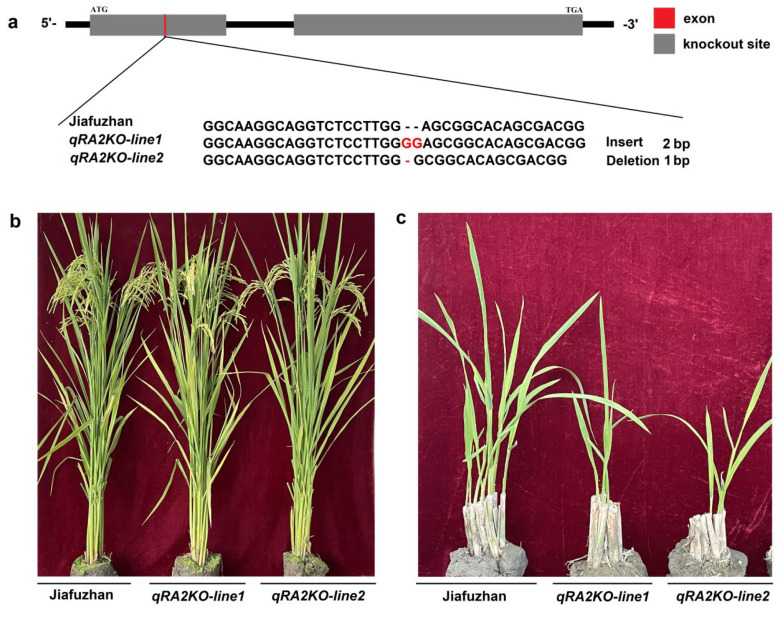
*qRA2*−*knockout lines* generated using CRISPR/Cas9 technology and exhibiting the NIL128 phenotype. (**a**): Two independent events (designated *qPA2KO*−*line1* and *qPA2KO*−*line2*) were generated using the CRISPR/Cas9 system and verified by sequencing. (**b**): Plant phenotype of Jiafuzhan and two knockout lines. (**c**): Ratooning ability comparison of Jiafuzhan and two knockout lines.

**Figure 4 ijms-24-00967-f004:**
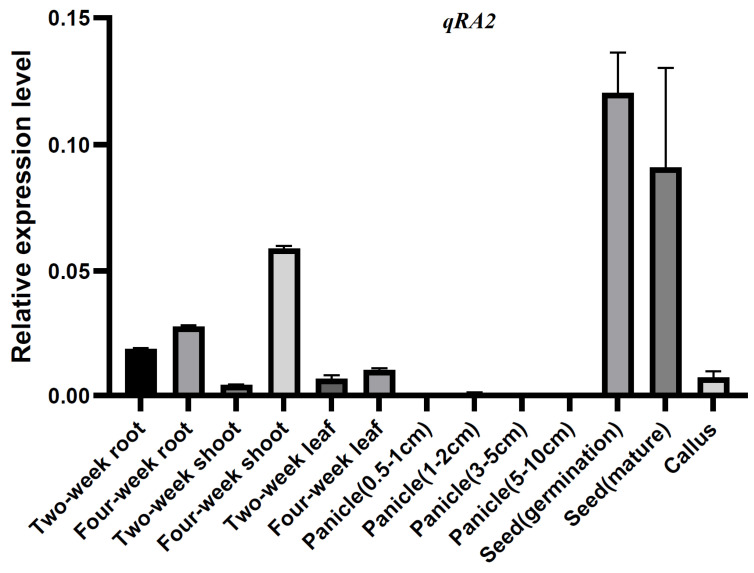
Determination of the expression patterns of *qRA2* via qRT−PCR. RNA samples were extracted from different tissues of Jiafuzhan, including roots, shoots, and leaves of two-, four- and six-week-old seedlings, spikelets of 0.5–1 cm, 1–3 cm, 3–5 cm, and 5–10 cm in length, germinating and mature seeds, and calli. Data represent the mean and standard deviation of three biological replicates. Three technical replicates for each biological sample were used. The error bar represents the standard deviation (SD) of the value from three independent biological samples.

**Table 1 ijms-24-00967-t001:** Comparison of the main agronomical traits of Jiafuzhan, NIL128, and the *qRA2KO* knockout mutant lines.

Traits	Jiafuzhan	Hui1586	NIL128	*qRA2* *KO-Line1*	*qRA2* *KO-Line2*
Plant height (cm)	118.32 ± 3.21	86.46 ± 1.94 **	119.43 ± 3.32	118.36 ± 3.21	119.28 ± 3.42
Panicle length (cm)	28.12 ± 1.23	29.35 ± 1.16 *	29.12 ± 1.31	27.82 ± 1.26	28.22 ± 1.41
Number of effective panicle	9.84 ± 1.04	10.06 ± 1.18	9.75 ± 1.02	10.01 ± 1.02	9.92 ± 1.09
Spikelets per panicle	163.26 ± 4.56	145.46 ± 3.88 *	164.12 ± 4.21	163.41 ± 4.32	164.46 ± 4.16
Seed setting rate (%)	92.52 ± 1.16	93.74 ± 1.66	86.22 ± 1.19 *	91.52 ± 1.26	93.12 ± 1.13
1000-grain weight (g)	23.12 ± 0.56	26.12 ± 0.54 *	23.22 ± 0.38	23.42 ± 0.61	23.36 ± 0.52
Grain length (mm)	10.84 ± 0.12	8.38 ± 0.22 *	10.72 ± 0.14	10.88 ± 0.15	10.88 ± 0.14
Grain width (mm)	2.62 ± 0.07	3.74 ± 0.08 **	2.64 ± 0.06	2.62 ± 0.09	2.60 ± 0.08
Yield per plant (g)	34.36 ± 0.92	35.82 ± 1.01 *	32.04 ± 0.98 *	35.06 ± 1.02	35.49 ± 0.94
Ratooning ability (%)	137.5 ± 5.85	52.3 ± 4.92 **	49.2 ± 4.83 **	46.8 ± 5.22 **	47.2 ± 4.96 **

* and ** indicate the significance levels of the differences between Jiafuzhan, NIL128 and knockout lines at *p* < 0.05 and *p* < 0.01, respectively. The data was derived from the trial that was performed at the Fuzhou experimental station in June 2021.

**Table 2 ijms-24-00967-t002:** Segregations of F_2_ populations crossed by NIL128.

Crosses	F_1_ Phenotype	F_2_ Population			*χ*^2^ (3:1)	*p*
		The Phenotype of Jiafuzhan	The Phenotype of NIL128	Total Plants		
NIL128/Jiafuzhan	The phenotype of Jiafuzhan	988	325	1313	0.336 *	>0.9

* Denote the segregation ratio of normal plants to mutant plants complied with 3:1 at 0.05 significant probability level. The data was derived from the trial that was performed at the Sanya Experimental Station in April 2020 and at the Fuzhou Experimental Station in August 2020.

**Table 3 ijms-24-00967-t003:** Indel and SSR molecular markers used for fine mapping of *qra2*.

Marker	Sequence of Forward Primer	Sequence of Reverse Primer
RM7286	CAGAACAATTCGACCGCTTC	GGCTTGAGAGCGTTTGTAGG
RM1386	CTACTCCCTAGTTGGCAGCG	TCTCCTGCAGGTACGTGCC
RM3795	CATTTGCATGGAGAGGATAG	TCATCTTCATTTCATTTCACC
Indle2-1	GTCTTGGAATTAAATGCTGC	GACCAAGATAAATGACAGGC
Indle2-3	TACAGGTTAAGAGGAGGCAA	CAGTTGCAGGATTTATCTGT
Indle2-7	AAGATATATAAACGCGTCGG	ATTGTCCTAAACGTACTGCC
Indle2-11	TCCATTGTCTTTCATCCTCT	GTGGTACCGATCAATTTCAG
Indle2-12	TCTTTTGACAATTTCCCATT	CGCAATGACCTTATCTGATT
Indle2-15	AGACGATGCGTAGACAAGAT	TGTATCGTCGTCTTTTGTTG
Indle2-17	TGAGACTAAGCGAAGGTAGG	CTCAAGCTTACAATTGACCC
Indle2-20	TTTCTCCTGAATGAATTGCT	CCATCCTTTCTTCTGGTACA
Indle2-24	TCACTACACAAACATACAAC	GCATGTCAACTAAATGGGTT
Indle2-28	TATTCGGCCTCTATATCCAA	CTTTCTCTAATAAAAAGATA
Indle2-33	CTTCTTCGACGACCTGGG	CAATTTCCAAGCTCTCTCC
Indle2-35	AGATATCGCAAGTTAGGCTG	TAGGGCAAAAGTTAAAACGA
Indle2-38	TTCGATCCTTTTTAATCACC	CACATGTTGTGTTGTAGATG
Indle2-40	TTTGAAGCTATATGCGTCTG	ATGTGTGCCATGATTTACAA

**Table 4 ijms-24-00967-t004:** Mutation site of three target mutant lines.

Line	Target Type	Mutation Site
*qRA2KO-line1*	gRNAs	GGCAAGGCAGGTCTCCTTGGGGAGCGGCACAGCGAC (Insert 2 bp)
*qRA2KO-line2*	gRNAs	GGCAAGGCAGGTCTCCTTGG----GCGGCACAGCGACGG (Deletion 1 bp)

## Data Availability

All data generated during this study are included in this published article and its Appendix A, and the raw data used or analyzed during the current study are available from the corresponding author upon reasonable request.

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
