# Peer review of "Fine Mapping and Cloning of a qRA2 Affect the Ratooning Ability in Rice (Oryza sativa L.)"

_ijms, 2023, doi:10.3390/ijms24020967_

Round 1

Reviewer 1 Report

Dear Prof. Dr. Editor of the International Journal of Molecular Sciences,

I write you regarding Manuscript ID: ijms-2065722 entitled " Fine mapping and cloning of a QTL, qRA2, which affects the ratooning ability in rice (Oryza sativa L.)" which was submitted to the International Journal of Molecular Sciences.

In this manuscript, the authors studied the Fine mapping and cloning of a QTL, qRA2, which affects the ratooning ability in rice (Oryza sativa L.). This work was well done and appropriate. The manuscript is suitable for publication in the International Journal of Molecular Sciences.

I have gone through this work. My decision is accepted with minor revisions for this work. The reason for that is as follows:

The manuscript deals with " Fine mapping and cloning of a QTL, qRA2, which affects the ratooning ability in rice (Oryza sativa L.)".

First: Title: It should change to the following:

Fine mapping and cloning of a qRA2 affect ratooning ability in rice (Oryza sativa L.)

Second Abstract, keywords and Introduction:

2) has some minor corrections.

Third: The other parts

3) has some minor corrections.

References

4) The references are ok.

Thank you for suggesting me as a reviewer for this manuscript.

with best regards

Author Response

Reviewer 1

I write you regarding Manuscript ID: ijms-2065722 entitled " Fine mapping and cloning of a QTL, qRA2, which affects the ratooning ability in rice (Oryza sativa L.)" which was submitted to the International Journal of Molecular Sciences.

In this manuscript, the authors studied the Fine mapping and cloning of a QTL, qRA2, which affects the ratooning ability in rice (Oryza sativa L.). This work was well done and appropriate. The manuscript is suitable for publication in the International Journal of Molecular Sciences.

I have gone through this work. My decision is accepted with minor revisions for this work. The reason for that is as follows:

The manuscript deals with " Fine mapping and cloning of a QTL, qRA2, which affects the ratooning ability in rice (Oryza sativa L.)".

Response: Thank you for your patient and meticulous review.

First: Title: It should change to the following:Fine mapping and cloning of a qRA2 affect ratooning ability in rice (Oryza sativa L.)

Response: This is a very good point. We have changed “Fine mapping and cloning of a QTL, qRA2, which affects the ratooning ability in rice (Oryza sativa L.)” to “Fine mapping and cloning of a qRA2 affect ratooning ability in rice (Oryza sativa L.)”.

Second Abstract, keywords and Introduction: has some minor corrections.

Response:  Thanks for the suggestions.  We have modified and improved it according to the annotation requirements in the abstract keywords and introduction part.

Third: The other parts: has some minor corrections.

 Response:  Thanks for the suggestions. We have modified and improved it according to the annotation requirements in other parts.

References:The references are ok.

Response:  Thanks.

Reviewer 2 Report

The presented manuscript has good potential but the authors need to address the following points. 

The introduction needs to be restructured as it is too difficult to get what is the main problem and what achieving this will solve for the rice community. 

The material and methods section is very weak and I will suggest authors to provide detail on how the used markers were selected. Provide which test was used to select the distorted SNP markers, and also, present the recommendation fraction across the different linkage groups to see the marker order on each bin (this will have implications on the  "Molecular Mapping of QTLs for Ratooning Ability, Physical Map Construction and Bioinformatics Analysis of QTLs for Ratooning Ability and Targeted Mutagenesis of qRA2 in Rice with CRISPR/Cas9"

The result section will be justified based on these new materials and methods. The data collected on the rationing was not well presented (year, location, .....) please clarify this. 

Author Response

Reviewer 2

The presented manuscript has good potential but the authors need to address the following points. 

Response: Thank you for your patient and meticulous review.

1)The introduction needs to be restructured as it is too difficult to get what is the main problem and what achieving this will solve for the rice community. 

Response:  Thank you for your good advice. We have added to the research on what the main problems are and what problems achieving this goal will solve for rice development (Line 58-64).

2)The material and methods section is very weak and I will suggest authors to provide detail on how the used markers were selected. Provide which test was used to select the distorted SNP markers, and also, present the recommendation fraction across the different linkage groups to see the marker order on each bin (this will have implications on the  "Molecular Mapping of QTLs for Ratooning Ability, Physical Map Construction and Bioinformatics Analysis of QTLs for Ratooning Ability and Targeted Mutagenesis of qRA2 in Rice with CRISPR/Cas9" 

Response:  Thanks for the suggestion.  a) We have provided detail on how the used markers were selected (Line 310-314). b) In this study, SNP markers are generally the difference changes of a single base, with relatively high polymorphism and a high number of markers, but only after enzyme digestion can the polymorphism between them be detected. Therefore, the INDEL molecular marker developed in this study is relatively convenient to use, but the polymorphism is not high, and a certain number of markers need to be developed to meet the needs of this study. c) In the process of fine localization in this study, physical distance was used for each molecular marker, that is, the actual concrete on the rice chromosome corresponding to each molecular marker, not the relative distance. Therefore, the corresponding physical distance was marked on the top (Figure 2 a) or bottom (Figure 2 b) of each marker. In addition, in order to make the method more specific and clear, we added the specific methods and steps of how to obtain the physical distance of each molecular marker in the method (Line 318-323).

3)The result section will be justified based on these new materials and methods. The data collected on the rationing was not well presented (year, location, .....) please clarify this. 

 Response:  This is a very good point.  The results are partially supplemented and improved based on these new materials and methods (Line 132-134). We supplemented and modified the data collected by rationing, including year, location, etc. (Line 78-79, Line 92-93, Line 112-113, Line 180-181, Line 19, Line 292-296).

Round 2

Reviewer 2 Report

Authors should provide recombination fraction , and the genetic map.

Author Response

Reviewer 2

Authors should provide recombination fraction , and the genetic map.

Response: Thank you for your patient and meticulous review. We have provided the corresponding recombination fragment and corresponding physical map for each primer marker (Figure 2b and Line 139-140).